# A Three-Step Procedure for Emergency Cerclage: Gestational and Neonatal Outcomes

**DOI:** 10.3390/ijerph19052636

**Published:** 2022-02-24

**Authors:** Manuel Gómez-Castellano, Lorena Sabonet-Morente, Ernesto González-Mesa, Jesús S. Jiménez-López

**Affiliations:** 1Obstetrics and Gynecology, Materno-Infantil Hospital Regional Universitary Málaga, Avd Arroyo de los Angeles S/N, 29011 Malaga, Spain; manugocas@hotmail.com (M.G.-C.); egonzalezmesa@gmail.com (E.G.-M.); jesuss.jimenez.sspa@juntadeandalucia.es (J.S.J.-L.); 2Biochemistry and Immunology Department, Malaga Biomedical Research Institute—IBIMA, University of Málaga, Surgical Specialties, 29010 Malaga, Spain

**Keywords:** cervical incompetence, emergency cerclage, preterm birth

## Abstract

(1) Background: The objective of our prospective observational study was to evaluate a new technique for emergency cerclage, which was performed on a cohort of patients with cervical incompetence in the second trimester. (2) Methods: 26 pregnant women presented at 15 to 24 weeks of gestation with cervical dilatation and bursa prolapse. A new emergency cerclage was performed using a technique consisting of the first cerclage in a tobacco bag and a second occlusive cerclage located inferiorly to the first. The technique is concluded with the performance of a cervical cleisis when vaginal bag prolapse is present, and this overall procedure is called the three-step procedure for emergency cerclage (TSEC). (3) Results: To assess its effectiveness, we measured the latency from procedure to delivery, pregnancy duration, infant birth weight, and rate of premature amniorrhexis. The mean latency from procedure to delivery was 14 weeks + 6 days, the mean weight of newborns was 2550 g and the mean gestational age at delivery was 35 weeks. The neonatal survival rate was 95.8%. The rate of premature amniorrhexis (<34 weeks gestational age) was 8.3% (two cases) with successful perinatal outcomes. There were significant differences (*p* < 0.05) between groups. A multivariate regression model showed that the best variables for predicting the latency to delivery were the cervical dilatation at diagnosis, use of the three-step cerclage, cervical length after the procedure, and gestational age at diagnosis. (4) Conclusions: The excellent results obtained with the TSEC procedure in terms of the latency from the procedure to delivery, gestational age at delivery, birth weight, and having few reported complications highlight the importance of collecting new data on this promising novel procedure.

## 1. Introduction

Preterm delivery accounts for 70–80% of neonatal mortality, and the associated intensive care, support after hospital discharge, and care during childhood impose great costs on the health systems of developed countries [1,2,3,4]. Women who present with cervical shortening are considered at high risk of preterm delivery [5]. Although the incidence of cervical insufficiency is difficult to establish due to the lack of clear diagnostic criteria, this pathology could be responsible for approximately 10–25% of gestational losses in the second trimester [6].

Cervical insufficiency is defined as an open cervical ostium and bulging amniotic membranes, without pain or other symptoms of threatened abortion (e.g., external bleeding or uterine contractions) [1]. Multiple observational studies have highlighted the efficacy of emergency cerclage in these patients, evidenced by the prolongation of pregnancy duration. Both a bibliographic review published by Namouz [7] and a recent meta-analysis by Chatzakis [8] concluded that emergency cerclage in pregnant women experiencing painless cervical dilatation decreases the likelihood of preterm birth, prolongs pregnancy, and reduces neonatal deaths and fetal losses without increasing the risks of chorioamnionitis and preterm premature membrane rupture.

Various techniques have been described for performing cerclage, including those described by Shirodkar [9] and McDonald [10] in the 1950s, which are still performed today. Shirodkar initially introduced a transvaginal cervical cerclage to be performed after detection of a gradually yielding cervix, even on women in the seventh month of pregnancy. McDonald performed transvaginal cerclage on women with cervical dilatation and bulging amniotic membranes during the second trimester of pregnancy [10]. Throughout the years, modifications to the original techniques have been introduced [11], including in terms of the level of the cerclage [12] and the suture materials to be used [13].

There is considerable controversy regarding the cerclage success rate and maternal–fetal outcomes. Bayrak et al. [14] reported that in 27 patients who received an emergency McDonald cerclage, 63% reached week 28 with an average prolongation in pregnancy of 64 days. Curti et al. [15] reported on 37 women between 17 and 27 weeks’ gestation with cervical dilatation who received a Shirodkar cerclage, where the pregnancy was prolonged for an average of 43 days and the average gestational age at delivery was 29 weeks. Studies have been carried out to compare the benefits of various techniques, but no technique has been found to be superior to the others. Basbug, in a recent study, compared the efficacies of the modified Shirodkar and McDonald rescue cerclage techniques in women with singleton pregnancies. The study sample included 47 women who presented with cervical incompetence and cervical dilatation with fetal membranes prolapsed into the vagina. The outcomes were compared according to the cerclage technique used: Shirodkar or McDonald. Both groups had similar delivery rates after 28, 32, and 37 weeks, and the researchers concluded that the McDonald and modified Shirodkar cerclage procedures had similar effects on prolonging pregnancy and improving the live birth rate [16].

Following a review of the literature on perinatal outcomes in patients who required emergency cerclage, we found that despite the application of treatments, the rates of prematurity and related complications remain high, and none of the cerclage techniques described so far were shown to be superior to the others. This is why we see the need to continue research into new technical modifications that can improve these results.

This study aimed to evaluate pregnancy outcomes in pregnant patients with cervical dilatation and prolapsed fetal membranes in the second trimester of pregnancy who were treated using a novel three-step procedure for emergency cerclage (TSEC) developed at our hospital.

## 2. Materials and Methods

During 2015–2020, all pregnant women between 15 and 24 weeks of gestation who presented with clinical cervical modification with or without membrane prolapse, or who were diagnosed with this condition after ultrasound diagnosis, were offered the option of undergoing emergency cerclage to prevent preterm delivery. This procedure was offered only when one of the following clinical criteria was met: (1) cervical length < 15 mm, measured using a 2.5–6.0 MHz transvaginal ultrasound probe (measurement had to be taken in the lithotomy position with the bladder empty, taking the shortest recorded measurement as the length) or (2) examination-based finding of 1–4 cm cervical dilatation with visualization of the membranes at or exceeding the level of the external cervical os. The cerclage was not performed in the presence of fetal anomalies, bacterial vaginosis, uterine contractions, preterm premature rupture of membranes, active labor, or when there were clinical symptoms or laboratory findings that suggested chorioamnionitis.

Prior to surgery, all the patients underwent a 24 h observation period to rule out chorioamnionitis, bacterial vaginosis, and active labor. In addition, vaginal ovules with chlorhexidine were applied 24 h before the operation.

Premature rupture of membranes (PROM) was determined using Actim^®^ PROM (Medix Biochemica, Espoo, Finland) or by the direct identification of amniotic fluid through the cervix and sonographic characteristics of oligohydramnios or polihydramnios. Chorioamnionitis was defined by a fever (>38 °C), significantly elevated maternal serum leukocyte count (>15,000/mm^3^) [11], and the combination of a positive amniotic Gram stain and glucose < 5 mg/dL. Bacterial vaginosis was diagnosed when purulent vaginal discharge was detected during the speculum examination on admission. Active labor was defined as regular uterine contractions, with three or more in 10 min, accompanied by cervical modifications.

Patients who underwent the procedure were informed that the surgeon could choose to make adaptations to the conventional cerclage techniques depending on the clinical conditions.

All patients provided the necessary consent to undergo cerclage and participate in the study. This study was approved by the Ethical Committee of the Hospital Regional Universitario de Málaga.

### 2.1. Operative Procedure

After diagnosing cervical incompetence, our group modified the cerclage technique as follows:A first cerclage in a purse-string suture to ensure bag reduction.Subsequent occlusive cerclage to ensure good cervical competence.Posterior cervical cleisis is optionally added depending on the cervical dilation and degree of prolapse (resulting in TSEC).

The TSEC was reserved for patients with dilation exceeding 3 cm and the presence of a prolapsed bag, while the McDonald-type cerclage was performed on patients with a greater cervical length and less dilation, and we reserved the double cerclage without cleisis for the remaining patients.

The TSEC was performed as follows:The cervix was exposed using Sim’s specula, after initial washing and asepsis of the vagina with chlorhexidine.The bag was reduced using a swab impregnated with sterile lubricant, until both cervical lips were exposed (the lower lip normally proves to be more difficult), and these were then pulled up using a Foerster clamp.A Foley catheter was prepared and cut at the distal end, at the level of the upper edge of the inflated balloon. Care was taken to not leave any edges that could damage the amniotic sac.The Foley catheter was inserted and then filled depending on the degree of cervical dilation and the stage of the procedure. For this, a third assistant helped to increase or reduce the drainage flow, depending on the stage of the procedure, which initially corrected the prolapse. The drainage flow was then reduced to cross the cervical canal, and later, once past the internal cervical os, the volume was increased again to reduce the bag and facilitate safe cerclage (Figure 1).The first cerclage was performed using Prolene 1. A purse-string suture was placed as cranially as possible and as close as possible to the level of the internal cervical os, with care taken not to damage the bladder. The suture was superficially applied without going too deep into the cervical stroma since this step aims to keep the bag reduced once the Foley catheter had been removed and to leave a segment of the cervix free on which to perform a second cerclage such that conglutination is guaranteed (Figure 2).The tobacco pouch seam was then closed while the Foley catheter was simultaneously deflated and removed, ensuring that the cervix was completely closed.A second cerclage was performed with Mersilene tape, which was attached approximately 1 cm below the previous cerclage. The stitches in this suture were designed to conglutinate the cervix, including the anterior and posterior lips at both commissures (8 to 11 o’clock on the left edge and 4 to 1 o’clock on the right edge). The point applied on the lower edge extended from 7 to 5 o’clock and was positioned as cranially as possible. The knot was located at 12 o’clock (Figure 3).The cervix was closed at the cervical os using Vycril 0. Two double stitches were applied in both commissures (Figure 4), plus a third in the central area (Figure 5).

Once the cerclage was performed, we evaluated whether it was correctly applied according to ultrasonographic visualization of the suture location, correct reduction of the pocket, and verification of a cervical length of >20 mm (Figure 6).

All patients were intravenously administered ceftriaxone in the operating room. Patients with bulging membranes at diagnosis were given additional prophylactic erythromycin and ampicillin IV during the first 48 h after the procedure.

Prophylactic tocolysis was indicated with 50 mg transrectal indomethacin every 6 h during the first 48 h, keeping the patient in the Trendelenburg position for the first 24–48 h. The patients were discharged after 72 h. Before hospital discharge, an ultrasound reevaluation of the cervix was performed to confirm the correct placement of the cerclage and the absence of cervical dilation or bag prolapse.

Cerclage removal was performed on an outpatient basis by sectioning the knots at week 37 if labor began, or when any circumstance that required termination of pregnancy occurred.

The main outcome measure was the mean latency from the placement of the different cerclages until delivery, as well as perinatal outcomes. The latency period was defined as the time elapsed from the application of cerclage to delivery. Other main outcomes were immediate maternal complications, including membrane rupture, pregnancy loss, excessive blood loss during the procedure (more than 25 mL), or cervical injury.

Further evaluated outcomes were the gestational age at delivery, the time elapsed from the application of cerclage, birth weight, and neonatal outcomes.

### 2.2. Statistical Analyses

We performed an initial analysis of the frequency distribution of the independent variables. Subsequently, a bivariate analysis was performed to identify associations between variables. For the bivariate analysis, we used an independent samples t-test to compare the mean values in two groups/categories of women when there was normality and a Mann Whitney U test in the remaining cases. For comparisons between a greater number of groups, we used either a single-factor ANOVA or the non-parametric Kruskal–Wallis test according to the conditions of homoscedasticity, which were evaluated using Levene’s test. The chi-squared test was used to compare qualitative variables. To analyze the relationships between quantitative variables, Pearson’s correlation coefficient was used. The significance level was set at *p* < 0.05. We used logistic regression models to predict the outcomes for the main dependent variable, i.e., the latency to delivery. The models were constructed using the Intro procedure, including the sociodemographic and obstetric variables that were first shown to be significantly associated using the typical stopping *p*-value thresholds for explanatory modeling [17].

## 3. Results

A total of 26 women between 15 and 24 weeks’ gestation underwent cerclage. Of these 26, 24 underwent TSEC, where the decision to perform cervical cleisis in 11 of these patients was based on their advanced cervical modification and the degree of amniotic membrane prolapse. The remaining two patients, with the most favorable clinical findings, underwent a McDonalds-type cerclage and were excluded from the study.

The mean age of our patients was 33.3 years. Of our sample, 54.2% had no previous delivery, and the remaining 45.8% were multiparas (Table 1).

The mean gestational age at diagnosis was 20 weeks + 1 day (±3 weeks + 4 days). The mean cervical length at diagnosis was 10.83 ± 7.92 mm and the mean cervical dilatation was 2.57 cm. After cerclage, the mean cervical length was 22.85 ± 8.23 mm.

Twenty-one patients (87.5%) presented with bulging amniotic membranes. In 14 patients, the membranes did not extend beyond the external cervical os, and in seven cases, the membranes were completely prolapsed into the vagina. Despite this, no case of accidental amniorrhexis occurred during surgery (Table 2).

After the intervention, four cases of chorioamnionitis were diagnosed, which represented 16.4% of the study cohort. Within this group, one case of stillbirth was recorded at week 24 of pregnancy, which occurred 4 weeks after cerclage. The woman had a satisfactory further evolution without adverse events. In the rest of the cases in which chorioamnionitis developed, the evolution of the newborn and the mother were satisfactory, with no infectious pathology after delivery or subsequent sequelae, leading to a disease-free survival of 75% for the newborns and 100% for mothers.

The prematurity rate (defined as birth before 37 weeks) was 54.2% (Table 3).

The rate of premature amniorrhexis < 34 weeks was 8.3% (two cases) with successful perinatal outcomes.

Table 4 shows the latency period to delivery, as well as the increase in the gestational age at delivery.

The cesarean delivery rate was 20.8% (five cases) and the vaginal delivery rate was 79.2%. There were no cases of cervical dystocia due to scar tissue that prevented cervical dilation.

We observed significant positive correlations between the cervical length after the intervention and gestational age (g.a.) at delivery, and between the cervical length at diagnosis and latency, with r = 0.48 (*p* < 0.05). On the other hand, significant negative correlations were found between both the dilatation at diagnosis and delivery and the latency duration, with r = −0.68 (*p* < 0.001).

In the bivariate analyses, there was a significantly shorter mean duration of pregnancy in women with chorioamnionitis than those without. The median interval to delivery was 7 weeks + 1 day in women with chorioamnionitis and 16 weeks + 1 day in women without (*p* < 0.007). A similar finding was revealed when the gestational weeks at delivery were compared between the two groups of women, with a significantly shorter median duration of pregnancy in the group with chorioamnionitis (28 weeks + 1 day vs. 36 weeks + 3 days; *p* < 0.003). The median neonatal birth weight in the chorioamnionitis group was 1285 ± 327.26 g, compared to 2803 ± 815.50 g in the control group, and this difference was significant (*p* < 0.002; Table 5).

Women with membranes bulging beyond the external os (12.5%) showed a mean latency to delivery after cerclage placement of 5 weeks + 6 days compared to 16 weeks in the control group (*p* < 0.01; dof 27.2), and the median neonatal birth weight was 1376 g compared to 2717 g for the control group (*p* < 0.01; dof 423.71; Table 5).

Diagnosis occurred significantly earlier in patients with a previous preterm birth (19 weeks vs. 20 weeks + 3 days (t = 0.79; *p* < 0.01; dof = 22)). In addition, the mean gestation at delivery was 36 weeks in patients with a previous preterm birth compared to 34 weeks + 5 days in the control group (*p* < 0.05; Table 5).

When we compared patients with a previous conization with those without, women with previous conization showed a longer interval to delivery (24 weeks vs. 13 weeks + 3 days (t = −2.8; *p* < 0.007; dof = −117.18)). Similarly, women with previous conization had a longer total gestation compared to women with no conization (38 weeks + 2 days vs. 34 weeks + 3 days (t = −1.3; *p* < 0.001; dof 21.49); Table 5).

After multivariable analysis, the best predictive fitting model for predicting the latency to delivery included cervical dilatation, cervical length after the intervention, technique for the cerclage, and gestational age at diagnosis (Table 6, Figure 7).

## 4. Discussion

To date, the treatments used for patients with cervical insufficiency diagnosed in the second trimester have failed to prevent a high rate of premature birth with its attendant neonatal consequences. Moreover, no technique for emergency cerclage had been shown to be superior to others.

In the present study, we analyzed the results of emergency cerclages performed on patients in the second trimester of pregnancy treated at our hospital using a cerclage technique that included certain technical variants that we considered potentially beneficial for patients. We have called our novel technique, the “three-step procedure for emergency cerclage (TSEC)”.

Our main objective was to evaluate how the application of our novel cerclage technique could prolong the duration of pregnancy. The results obtained were favorable compared to those reflected in the literature, not only in relation to the prolongation of pregnancy but also in terms of variables such as the neonate birth weight and the rates of maternal and fetal complications.

In our trial, the mean gestational age at cerclage was 20 weeks + 1 day and the interval to delivery after cerclage was 14 weeks + 6 days. Fortner et al. [17] reported a latency rate of 5 weeks + 3 days. On the other hand, Basbug et al. [16], who performed cerclage with the Schirodkar technique, reported a latency until delivery of 11 weeks + 3 days. Daskalakis et al. [18] reported on 46 patients with bulging fetal membranes. Seventeen patients were treated with bed rest and 29 were treated with McDonald’s cerclage, and an increase in the median prolongation of pregnancy of 8.8 weeks was observed in the group who underwent emergency cerclage. Ventolini et al. [19] enrolled 56 women who underwent Schirodkar cerclage. The mean gestational age at cerclage was 19 weeks + 4 days and the latency to delivery was 9 weeks + 1 day.

Our rate of preterm amniorrhexis in the pregnancies where cerclage was applied was 8.2%, which was much lower than rates of around 25% reported in the existing literature [20]. Stupin et al. [21] performed double cerclage plus type cleisis (Szendi/Saling) as well as McDonalds-type cerclage in their sample, and they reported a total of eight patients in whom amniorrhexis occurred during the procedure as well as three cases of cervical lacerations during the removal of the cerclage. In our study, there were no cases of accidental amniorrhexis during the procedure or any notable complications in its removal, nor did we observe any influence on the evolution of the subsequent delivery.

Gupta et al. [22] reported a chorioamnionitis rate of 46% and Abo-Yaoub et al. [23] reported a rate of 16.2%. Likewise, Freire et al. [24] reported a rate of 23.5%. Our chorioamnionitis rate was 16%, which was lower than that reported in other studies. Analysis of the cases where clinical chorioamnionitis developed revealed significant differences in terms of the duration of pregnancy (7 weeks + 1 day vs. 16 weeks + 2 days), indicating that this is the main factor determining cerclage outcome.

One of the most outstanding findings was that, in conization patients, the results of TSEC were very satisfactory, with the latency times to deliver even higher than those in non-conization patients. Even though our sample was limited, the differences found between the two groups were significant, and we suggest that this type of cerclage could be particularly beneficial for the group of conization patients.

It is especially important to evaluate how cerclage can prevent extreme prematurity. In his study, Aoki [25] succeeded in prolonging pregnancies beyond 28 weeks in 66% of patients. Bayrak [14] reported similar numbers (63%), as did Pereira (62%); [26,27]. In our study, 87.5% of the pregnant women who underwent cerclage managed to exceed 28 weeks of gestation.

There are very few reports on pregnancies that reach full-term. Ventollini [19] reported that 23% of pregnancies exceeded 38 weeks; however, Basbug [16] reported a lower figure of 13.6%. In our sample, 11 patients exceeded 37 weeks of gestation, which represents 45.8% of the total.

Birth weight is an important indicator of perinatal outcomes, and there are many reports on this in the literature. One study that stands out is that by Hordnes [28], which was published in 1996 and reported an average birth weight of 2252 g; however, the small number of patients in that study should be noted. As an example of a larger and more recent study, Fortner [19] reported an average birth weight of 1190 g, and in 2010, Gupta [22] reported an average birth weight of 1937 g. The results of our study were thus favorable since we found an average birth weight of 2550 g.

After analyzing all the above data and in accordance with the updated literature [29,30,31,32,33,34,35,36,37], we conclude that the use of our TSEC technique could extend the latency period and, accordingly, result in lower prematurity rates. Its effect was especially beneficial in patients with poor prognoses (prolapsed membranes, cervical dilatation, conization patients), for whom outcomes in most studies have been unsatisfactory. The TSEC is an easily applicable technique, and we find the low or null incidence of complications during its performance to be of particular interest. Together with the efficacy of the technique, this led to a drastic reduction in the lengths of the hospital stays of these patients, with a resulting improvement in cost-effectiveness.

## 5. Conclusions

In summary, cerclage is a technique with proven usefulness in treating patients with cervical dilation in the second trimester of pregnancy for preventing preterm delivery. It prolongs the pregnancy and prevents poor perinatal outcomes derived from prematurity.

To date, no specific cerclage technique has proven to be superior to the rest. In this study, we present results for our novel technique that introduces certain modifications—the TSEC.

Most of the studies to date, and the present study, report on a small number of recruited patients. This fact, together with the fact that most studies—including the present study—are observational, prevents us from drawing definitive conclusions. and represents the main limitations of this research, with larger studies required to confirm our findings.

Despite this, the impressive results in terms of the latency until delivery after cerclage, gestational age at delivery, birth weight, and minimization of complications reported in our work compel us to continue evaluating the use of our technique, which may significantly improve the outcomes for neonates and mothers.

## Figures and Tables

**Figure 1 ijerph-19-02636-f001:**
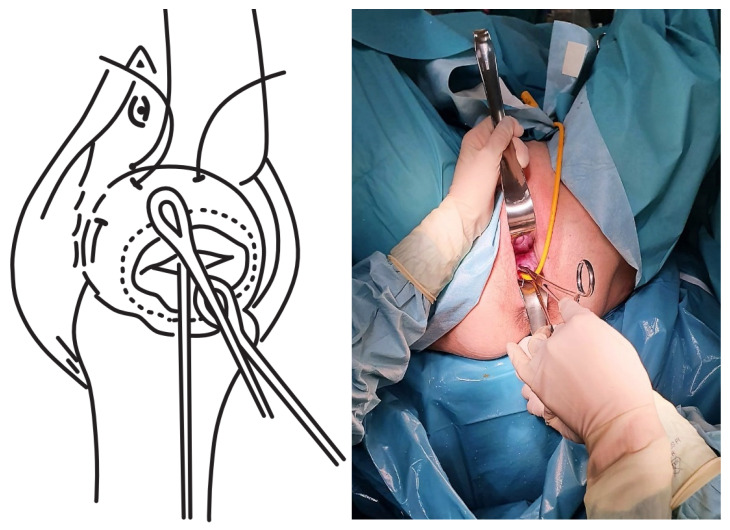
Reduction of prolapsed membranes.

**Figure 2 ijerph-19-02636-f002:**
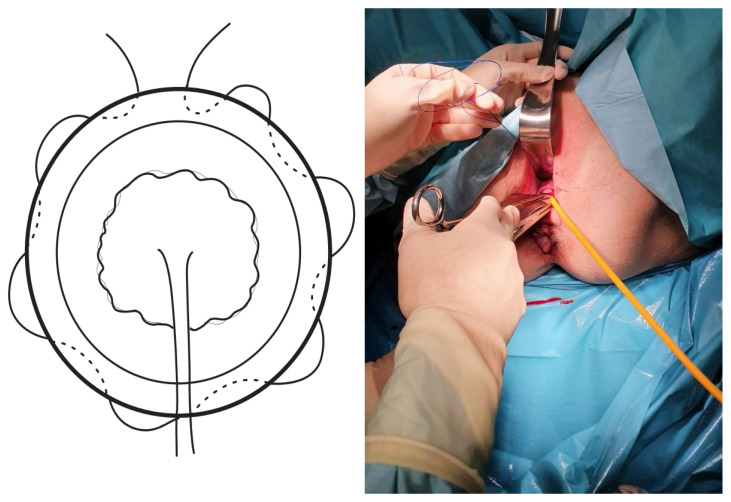
Purse-string sutures.

**Figure 3 ijerph-19-02636-f003:**
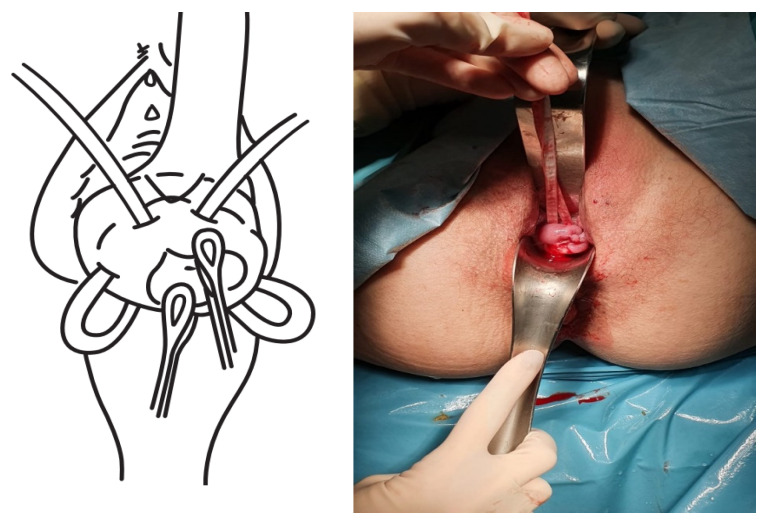
Second occlusive suture (Mersilene).

**Figure 4 ijerph-19-02636-f004:**
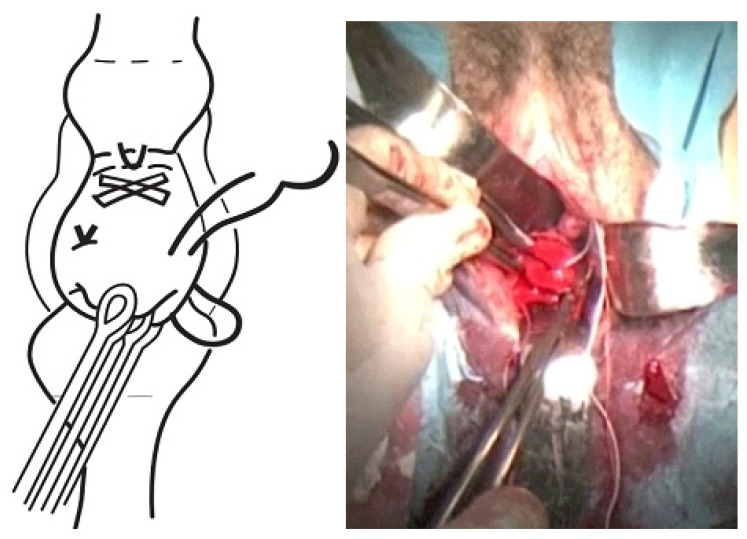
Commissures stitches.

**Figure 5 ijerph-19-02636-f005:**
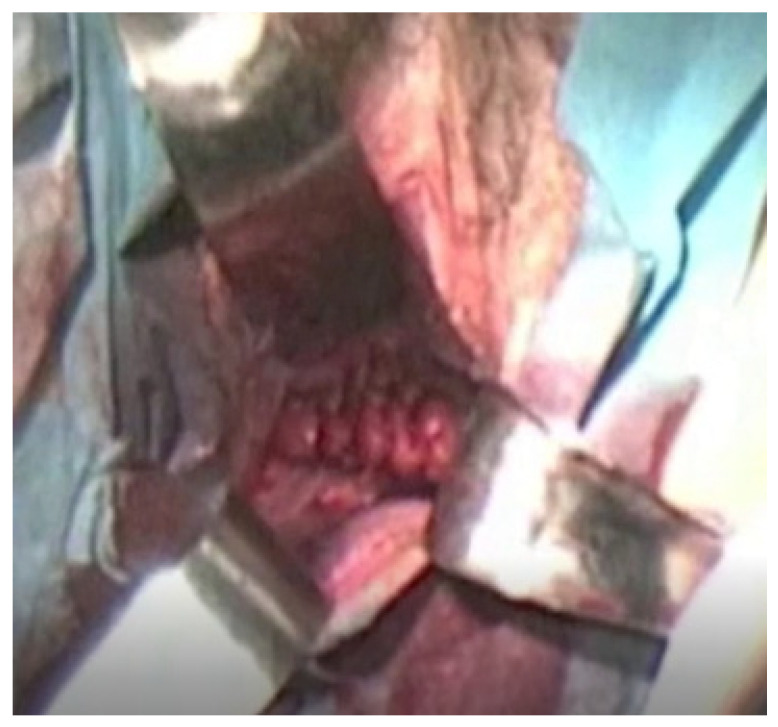
Cervical cleisis.

**Figure 6 ijerph-19-02636-f006:**
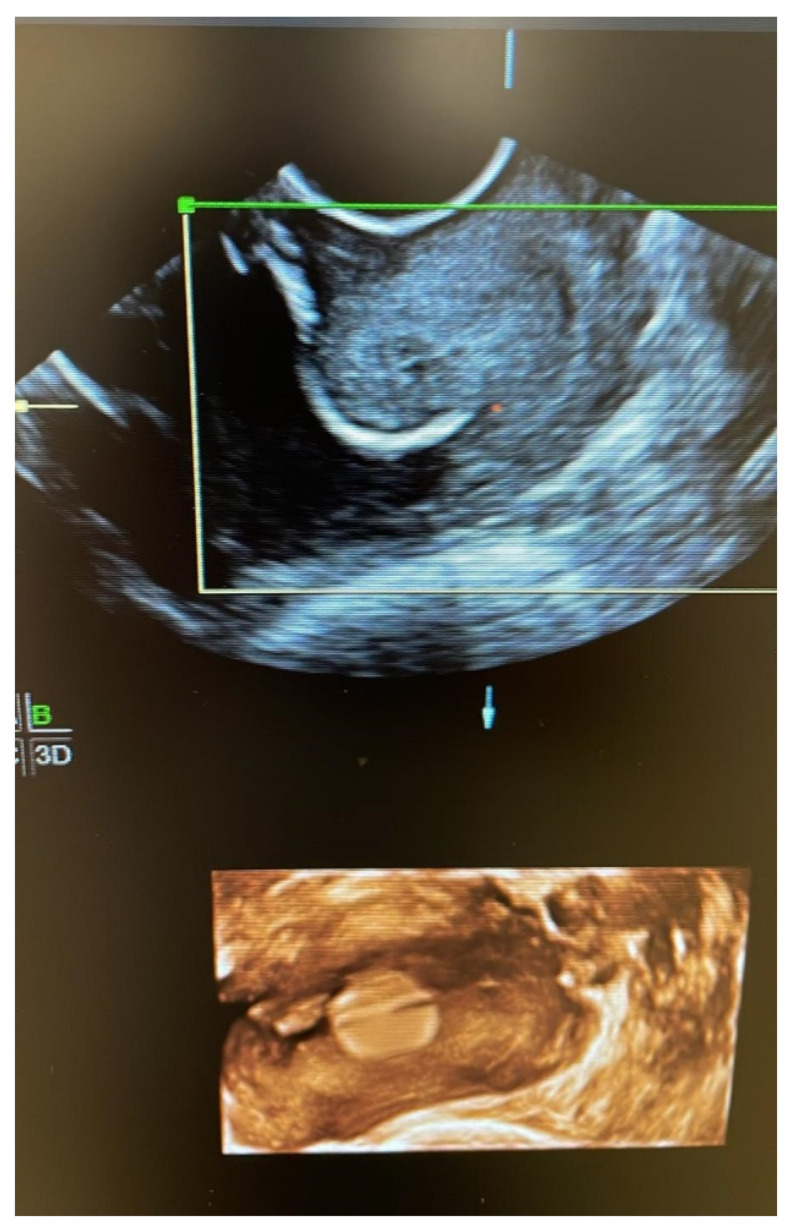
Three-dimensional sonographic cervical cerclage.

**Figure 7 ijerph-19-02636-f007:**
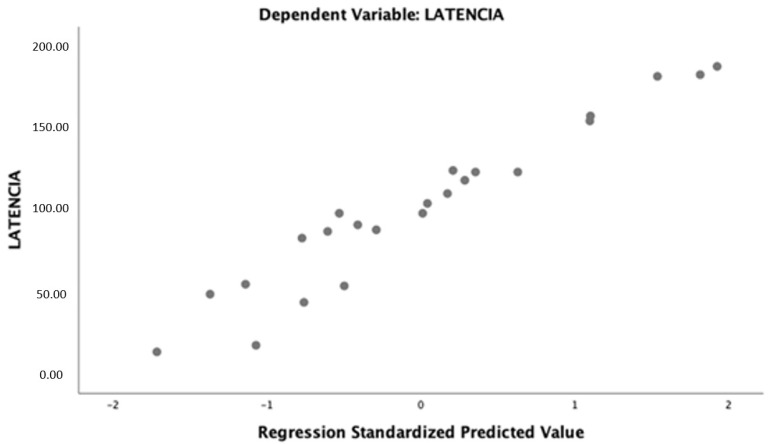
The best-fitting predictive model of logistic regression of the interval to delivery.

**Table 1 ijerph-19-02636-t001:** Demographic characteristics.

Variables	Patients (*N* = 24)
Age	33.5 ± 5.5
Parity	
Nulliparous/Primiparous	13 (54.2%)
Multiparous	11 (45.8%)

**Table 2 ijerph-19-02636-t002:** Cervical characteristics of our patients (*N* = 24).

	*N* (%)
Previous Abortion	1 (45.8%)
Previous Conization	3 (12.5%)
Previous Premature Delivery	5 (20.8%)
Previous Chorioamnionitis	2 (8.3%)
Previous Cervical Incompetence	3 (12.5%)
Previous Prolapsed Membranes	3 (12.5%)

**Table 3 ijerph-19-02636-t003:** Pregnancy Outcomes.

	*n* (%)
Chorioamnionitis	4 (16.4%)
Birth after 28 Weeks	21 (87.5%)
Birth after 34 Weeks	19 (79.1%)
Birth after 37 Weeks	11 (45.8%)
Neonatal Survival	23 (95.8%)

**Table 4 ijerph-19-02636-t004:** Duration of pregnancy and perinatal outcomes.

	Mean (Range)	SD
Gestation at Delivery (Weeks + Days)	35 (24 + 1 to 40 + 5)	±4 + 5
Interval to Delivery (Days)	14 + 6 (2 to 26 + 5)	±7
Birth Weight (g)	2550 (580–3880)	±947.266

Note: SD, standard deviation.

**Table 5 ijerph-19-02636-t005:** Bivariate analysis. Data are presented as median or mean ± standard deviation. R square 90% (*p* = 0.1).

	Chorioamnionitis			Preterm Birth			Conization			Prolapsed Membrane		
	+	−	*p*-Value	+	−	*p*-Value	+	−	*p*-Value	+	−	*p*-Value
Gestational at diagnosis, weeks + days	15 + 4	20 + 3	NS	19	20 + 3	*p* < 0.01	14 + 2	21 + 1	*p* < 0.007	17 + 4	20 + 3	NS
Gestation at delivery, weeks + days	28 + 1	36 + 3	*p* < 0.003	36	34 + 5	*p* < 0.05	38 + 2	34 + 3	*p* < 0.001	32 + 4	35 + 1	NS
Interval to delivery, weeks + days	7 + 1	16 + 1	*p* < 0.007	17	12	NS	24	13 + 3	NS	5 + 6	16	*p* < 0.01
Birth weight, g	1285 ± 327.3 g	2803 ± 815.5 g	*p* < 0.002	2628	2529	NS	2943	2493	NS	1376	2717	*p* < 0.01

Note: “+” and “−” refers to the presence or absence of the variable.

**Table 6 ijerph-19-02636-t006:** Multiple regression for latency.

Constant	B	Std Error	Standardized Beta Coefficients	t	Sig	95% Lower Bound	95% Upper Bound
Cervical Dilatation	−18.91	0.86	−0.49	−2.71	0.02	−34.27	−3.544
Three-Step Cerclage	43.782	24.073	0.185	1.819	0.096	−9.20	96.757
Length after Cerclage	2.110	0.655	0.360	3.22.	0.008	0.669	3.551
Gestational Days at Diagnosis	1.260	0.251	−0.660	5.024	0.000	1.812	−0.708

Dependent variable: interval to delivery.

## Data Availability

Information related to the cases can be obtained from the authors (M.G.-C. and L.S.-M.) on reasonable request. Data were collected during the patients’ hospital admission by the investigators, who provided direct assistance. These data are stored in the computerized Andalusian health support system of electronic medical records.

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
