# Peer review of "A Three-Step Procedure for Emergency Cerclage: Gestational and Neonatal Outcomes"

_ijerph, 2022, doi:10.3390/ijerph19052636_

Round 1
Reviewer 1 Report
My remarks
-
Reconsider the title: Gestational and Neonatal... What?
-
Were all pregnancies singleton?
-
Was diagnosis of bacterial vaginosis made by only by inspection? Were any swabs for bacterial analysis taken prior to surgery?
-
I understand that there are two assistants necessary for the procedure. Were all 22 TSECs performed by the same surgeon?
-
What happened if ultrasound after the procedure failed to confirm the appropriate placement of cerclage (≥ 20 mm)?
-
Table 1. Parity: Nulliparous and Multiparous should be together 100 % Or no?
-
Line 236: c. section 20.8 % and vaginal delivery 58.3%. What happened to the rest?
-
Chorionamnionitis has a deleterious effect on the outcome. Were there any common characteristics of the women that developed it? Any ideas how to prevent it?
-
Table 6.:Three-step cerclage as opposed to McDonald?
-
How do you explain better outcome in patients with previous conization?
-
Line 267: you probably mean gestational age at diagnosis?
Author Response
Dear reviewer, we appreciate your comments. We proceed to answer your suggestions one by one.
- Reconsider the title: Gestational and Neonatal... What?
A Three-Step Procedure for Emergency Cerclage: Gestational and Neonatal outcomes.
- Were all pregnancies singleton?
All pregnancies were singleton.
- Was diagnosis of bacterial vaginosis made by only by inspection?
Were any swabs for bacterial analysis taken prior to surgery? In all cases ,swabs for bacterial analysis were taken prior to surgery.
- I understand that there are two assistants necessary for the procedure. Were all 22 TSECs performed by the same surgeon?
The procedure was made by the differents surgeons of the emergency team.
- What happened if ultrasound after the procedure failed to confirm the appropriate placement of cerclage (≥ 20 mm)?
In all cases the cervical length,were > 10mm and all of them, the ultrasound confirm the correct placement of the cerclage (with a correct and functional closing of the internal cervical os) so,no further measure was taken.
- Table 1. Parity: Nulliparous and Multiparous should be together 100 % Or no? We have confirmed that there was a typographic mistake.
- Line 236: c. section 20.8 % and vaginal delivery 58.3%. What happened to the rest?
The rest of cases (20.9%) finalized by instrumental delivery.In our opinion we can add both percentage in the vaginal delivery group (79.2%).
- Chorionamnionitis has a deleterious effect on the outcome. Were there any common characteristics of the women that developed it? Any ideas how to prevent it?
Chorioamnionitis occurs in patients who presents a large bulged membranes at diagnosis. We suppose that it could be correlated with more exposition of membranes to a vaginal bacteries.
- Table 6.:Three-step cerclage as opposed to McDonald?
In this table we have made a multiple regression of the period of latency The contrast to the Mc Donald´s thecnique is not reflected in this analysis.
- How do you explain better outcome in patients with previous conization?
We suppose that the fact of those patients were considerated in a hight risk of preterm delivery, made that the pregnancy control in those cases was more often,and allowed us to make and earlier disgnosis of the problem.
- Line 267: you probably mean gestational age at diagnosis?
You are rigth. Actually we are refering to “gestational age at diagnosis”.
Reviewer 2 Report
I recommend that the authors differentiate patients with TSEC (with or without cleisis) and exclude from the study women who required McDonalds type cerclage.
It´s important that the results are based on the technique used (TSEC / Clveisis) and the situation at the time of surgery: dilatation and the degree amniotic membrane prolapse.
Author Response
Dear reviewer, we appreciate your comments. We proceed to answer your suggestions one by one.
I recommend that the authors differentiate patients with TSEC (with or without cleisis) and exclude from the study women who required McDonalds type cerclage.
We have revised our dates and the McDonald´s cerclajes were excluded.The total sample was 26,and the TSEC group was 24 patients.All the stadistics analyses,were made with the TSEC group. It´s neccesary to rewrite this paragraph: (line 205)
“A total of 26 women between 15 and 24 weeks’ gestation underwent cerclage. Of these 26, 24 underwent TSEC, where the decision to perform cervical cleisis in 11 of these patients was based on their advanced cervical modification and the degree of amniotic membrane prolapse. The remaining two patients, with the most favorable clinical findings, underwent a McDonalds-type cerclage.
It´s important that the results are based on the technique used (TSEC / Clveisis) and the situation at the time of surgery: dilatation and the degree amniotic membrane prolapse.
The cleisis ,effectively was performed in large polapsed membranes cases, and > 20mm cervical dilatation. We are recluiting more patients with those conditions at diagnosis to evaluate the efficiency of the cervical cleisis.
Reviewer 3 Report
A Three-Step Procedure for Emergency Cerclage: Gestational and Neonatal
I find the title „A Three-Step Procedure for Emergency Cerclage: Gestational and Neonatal” adequate considering the content of the article. The abstract section describes the author’s study purpose and the results. In the introduction section, the authors described the existing techniques for performing cerclage and the controversy regarding the success rate of the procedures. The authors also described the necessity to further research and improve cerclage techniques.
In the methods section, the three-step procedure is well detailed with figures and intraoperative photos. Each step is explained chronologically. The group of pregnant women was well described along with the admission criteria for the study. Also, the number of participants is not specified.
The results section reports data from all methods used. Results are statistically documented by text and tables. However, there are some observations to be made: - In table 1, the percentage of nulliparous (4.52%) doesn’t match with the information provided in the previous paragraph “[…] 54% had no previous delivery”; - The paragraph “The cesarean delivery rate was 20.8% (five cases) and the vaginal delivery rate was 58.3%” should be reviewed because an inconsistency in the data is observed.
In the discussion section, the authors described the results obtained by using the procedure, pointing favourable results compared to those described in the literature.
Although it seems to be a documented article, the authors should also cite more recent papers as some of bibliographic sources dates before 2010. Also, some minor spelling errors should be corrected (e.g line 241,320,333).
To conclude, although this is a well-documented manuscript, it needs minor revision before being accepted for publication.
Author Response
Dear reviewer, we appreciate your comments. We proceed to answer your suggestions one by one.
I find the title „A Three-Step Procedure for Emergency Cerclage: Gestational and Neonatal” adequate considering the content of the article. The abstract section describes the author’s study purpose and the results. In the introduction section, the authors described the existing techniques for performing cerclage and the controversy regarding the success rate of the procedures. The authors also described the necessity to further research and improve cerclage techniques.
In the methods section, the three-step procedure is well detailed with figures and intraoperative photos. Each step is explained chronologically. The group of pregnant women was well described along with the admission criteria for the study. Also, the number of participants is not specified.
The results section reports data from all methods used. Results are statistically documented by text and tables. However, there are some observations to be made: - In table 1, the percentage of nulliparous (4.52%) doesn’t match with the information provided in the previous paragraph “[…] 54% had no previous delivery”; - We have confirmed that there was a typographic mistake.
The paragraph “The cesarean delivery rate was 20.8% (five cases) and the vaginal delivery rate was 58.3%” should be reviewed because an inconsistency in the data is observed.
The (20.9%) finalized by instrumental delivery, for this reason , the data seems to be confused. We consider that we can add both percentage in the vaginal delivery group (79.2%).
In the discussion section, the authors described the results obtained by using the procedure, pointing favourable results compared to those described in the literature.
Although it seems to be a documented article, the authors should also cite more recent papers as some of bibliographic sources dates before 2010. Also, some minor spelling errors should be corrected (e.g line 241,320,333).
We hade used those studies in order to compare our work, with the largest sample studies as possible.
To conclude, although this is a well-documented manuscript, it needs minor revision before being accepted for publication.